# Integrating Biosignals Measurement in Virtual Reality Environments for Anxiety Detection

**DOI:** 10.3390/s20247088

**Published:** 2020-12-10

**Authors:** Livia Petrescu, Cătălin Petrescu, Oana Mitruț, Gabriela Moise, Alin Moldoveanu, Florica Moldoveanu, Marius Leordeanu

**Affiliations:** 1Faculty of Biology, University of Bucharest, 050095 Bucharest, Romania; livia.petrescu@bio.unibuc.ro; 2Faculty of Automatic Control and Computers, University Politehnica of Bucharest, 060042 Bucharest, Romania; catalin.petrescu@acse.pub.ro (C.P.); alin.moldoveanu@cs.pub.ro (A.M.); florica.moldoveanu@cs.pub.ro (F.M.); marius.leordeanu@cs.pub.ro (M.L.); 3Faculty of Letters and Sciences, Petroleum-Gas University of Ploiesti, 100680 Ploiesti, Romania; gmoise@upg-ploiesti.ro

**Keywords:** virtual reality, biophysical, emotion recognition, experiment, EDA, human-computer interaction

## Abstract

This paper proposes a protocol for the acquisition and processing of biophysical signals in virtual reality applications, particularly in phobia therapy experiments. This protocol aims to ensure that the measurement and processing phases are performed effectively, to obtain clean data that can be used to estimate the users’ anxiety levels. The protocol has been designed after analyzing the experimental data of seven subjects who have been exposed to heights in a virtual reality environment. The subjects’ level of anxiety has been estimated based on the real-time evaluation of a nonlinear function that has as parameters various features extracted from the biophysical signals. The highest classification accuracy was obtained using a combination of seven heart rate and electrodermal activity features in the time domain and frequency domain.

## 1. Introduction

The integration of biophysical signals recordings in immersive virtual reality (VR) applications and experiments may significantly improve the results of psychotherapy, as physiological responses proved to be more useful than subjective reports in estimating users’ emotions/anxiety levels during interactive sessions. By enhancing the human-machine interfaces and by augmenting the naturalness, fluidity, and intelligence of communication [1,2], we can facilitate the synchronization of physiological signals with VR scenario events to conduct a causality analysis.

Due to the complex and multidimensional nature of emotions, a holistic approach is needed to characterize the global psychophysiological responses using different evaluation systems. In the literature, there are two main methodological ways, namely: Techniques that use self-assessment questionnaires [3] and techniques that use measurements of various physiological parameters. These techniques can be used separately, but are often employed simultaneously to increase the reliability and quality of results [4]. The activity of the vegetative nervous system, responsible for the causality of emotions, is mostly evaluated by recording electrical impulses from the nervous system (electroencephalography—EEG) and by measuring various body parameters, such as electrodermal activity (EDA) and heart rate (HR).

The results of the simultaneous recording of these parameters can sometimes show variations in the same direction, but in other situations, they vary in opposite directions. These evaluation techniques address different emotion dimensions—arousal, valence, or dominance [5]. For instance, EDA is a sensitive indicator for lower arousal range, responsible especially for cognitive processing, while HR is an indicator of higher arousal range, being related especially to the somatic mechanism of arousal [6].

As VR techniques started to be used in biomedical applications relatively recently, there are still no standardized protocols regarding the special conditions for the acquisition and processing of biological signals. The existing protocols cannot always be successfully used in VR applications, due to particularities, such as the long duration of the acquisition session, the large volume of resulting data, or the numerous artifacts caused by the subject’s dynamics during the experiment.

This paper proposes a protocol, specifically designed to improve the biophysical signals’ acquisition and processing quality, which may estimate the subject’s anxiety level based on physiological measurements. This estimation can be used to adapt the phobia therapy sessions according to the subject’s emotional state. For a better understanding of the protocol, we propose a case study analysis.

Section 2 presents emotions and exposure therapy alternatives, Section 3 describes the virtual reality applications used in psychophysical experiments, Section 4 discusses the most relevant biophysical signals-electrodermal activity, heart rate, and electroencephalography, Section 5 presents various protocols for physiological data acquisition in VR, Section 6 brings forward the proposed methodology for physiological data acquisition in VR, while Section 7 describes the case study for acrophobia therapy. Finally, the results, discussions, and conclusions are exposed in Section 8, Section 9 and Section 10.

## 2. Emotions and Exposure Therapy

Emotion is a mental state that appears spontaneously and without conscious effort, reflected by physiological changes in the brain, heart, muscle, and tissue activity. Emotions have been classified using two models: The discrete model, which divides emotions into various categories, and the continuous model, where emotions are labeled using multiple dimensions. The discrete models emerged considering the idea that emotions are innate, common in similar situations, and manifested psychologically and physiologically with comparable patterns across all people.

Initially, Ekman [7] identified six basic emotions: Happiness, sadness, anger, fear, surprise, and disgust and considered that the other affective states result as a combination of these six basic emotions. A set of eight basic emotions has been proposed: Joy, trust, fear, surprise, sadness, disgust, anger, and anticipation [8] and later, a set of ten basic emotions has emerged: Surprise, joy, interest, sadness, fear, shyness, disgust, guilt, anger, and contempt [9].

The continuous model classifies emotions based on the arousal and valence dimensions [10]. Valence describes the quality of the emotion and ranges from positive (pleasant) to negative (unpleasant). Arousal characterizes the degree of physical and cognitive stimulation and ranges from low arousal (passive) to high arousal (active) [11]. Emotions are spanned across this 2D emotion space model. As various, very different emotions fall within the same quadrant in the 2D model, a third dimension has been proposed by Mehrabian [12]. Dominance reflects the ability to control emotions and ranges from submissive to dominant. As an example, both fear and anger belong to the same quadrant in the 2D space—low valence and high arousal. What differentiates them is the fact that anger is a dominant state, while fear is submissive (low dominance).

Fear and anxiety are emotions that allow an appropriate response to a dangerous situation or threat. Although both are alarm signals, each prepares the body for a different kind of response. Thus, fear is a response to external, current danger, while anxiety is a response to an imagined threat, or an internal conflict. It is important to emphasize the indispensable cognitive component of the anxiety reaction. Consequently, anxiety can be seen as a more elaborate form of the fear reaction, allowing us to adapt to potential reactions and plan for the future [13]. Phobic-anxiety disorders are a group of manifestations characterized by exaggerated, uncontrollable, disproportionate fear, and avoidant behavior. Phobic people can present both intense mental discomfort that can lead to panic attacks and physical discomfort, with palpitations, breathing disorders, generalized weakness, profuse sweating, dizziness, etc. It is important to make a difference between fear and phobia. Fear has an important adaptive role, being a transient state that appears in direct confrontation with danger, while a phobia is a pathological tonic state of preparation and continuous prediction [14].

VR exposure therapy has become an option for in vivo exposure and has several advantages: Authenticity of exposure, safe exposure, unlimited possibilities of training situations, high control of the variables for the clinician, and low costs. The field that examines the affective states is called affective computing. In 1997, Picard defined affective computing as: “Computing that relates to, arises from, or deliberately influence emotions” [15]. All the applications aimed at studying emotions and their psychological and physiological responses in education, healthcare, therapy, marketing, media, etc., are part of the broad field of affective computing.

## 3. Virtual Reality in Psychophysical Experiments

In psychophysical experiments, **emotion elicitation** has been performed using various stimulation materials: Pictures, music, short videos, games, virtual reality, self-induction methods. The International Affective Picture System (IAPS) [16] or the Chinese Affective Picture System (CAPS) [17] contain a set of standardized, open-access emotion-eliciting pictures available for a wide range of research experiments. Besides IAPS, the Center for the Study of Emotion and Attention from the University of Florida proposed the International Affective Digital Sounds [18] dataset, which contains auditory stimuli across a wide range of semantic categories [19]. The Affective Norms for English Words (ANEW) provides emotional ratings of valence, arousal, and dominance for a large number of English words [20]. The Affective Norms for English Text (ANET) contains emotional ratings of valence, arousal, and dominance for a large set of short English texts [21]. Music videos have been used for eliciting emotions in [22,23]. In [24], one actor tried to self-induce five major emotions: exhaltation, happiness, boredom, fear, and a neutral state.

The modalities of quantifying emotions are: Self-reports (verbal descriptions of the emotions experienced, self-ratings using the Self-Assessment Manikins (SAM)—various anthropomorphic figures [25] depicting different emotions, the Affective Slider [26]—a valid alternative to Self-Assessment Manikins, observations of the human behavior and measures of the physiological signals originating in the Central Nervous System (brain activity collected by EEG) and Autonomic Nervous System (respiration, heart rate, body temperature).

Virtual reality creates a complete sense of presence and increases immersivity by using stereoscopic 3D view, motion tracking, and head tracking technology [2]. It is a rapidly growing field, expected to rise to 80 billion dollars by 2025 [27]. Immersive Virtual Reality is able to trigger emotional states, quantified by various physiological responses.

Spatial presence is an important prerequisite of any VR application, describing the perception of being present in the virtual environment [28]. Spatial presence enhances game attractivity and improves players’ performance [29]. Felnhofer et al. [30] demonstrated that the sense of presence is linked to emotional responses in an experiment where electrodermal activity has been recorded.

The feeling of presence in the virtual environment, the degree of realism, and immersivity are highly correlated with the percentage of change in heart rate, and skin resistance. Wiederhold [31] found a significant correlation between presence and skin conductance level in VR. Riva et al. [32] found that the sense of presence influences the emotions experienced in the virtual environment. Jang et al. [33] found that different virtual environments generate different affective states. Moreover, electrodermal activity and heart rate variability are reliable measures of arousal. In Mehan et al. [34], the intensity perceived in the virtual environment was directly proportional to the heart rate. Peterson et al. [35] showed that high virtual height conditions increased heart rate variability, electrodermal activity, and heart rate frequency power compared to low virtual heights, inducing a certain level of psychological stress. Virtual characters, such as avatars, are capable of eliciting emotional responses in people who observe their facial expressions and postural gestures [36,37,38].

Virtual reality has been successfully used for phobia therapy. In [31], the post-therapy behavior of 33 phobic subjects who have been treated for fear of flying showed a gradual trend towards the nonphobics’ physiological responses, due to desensitization (the process that diminishes emotional responsiveness to negative or aversive stimuli) and habituation (the repeated presentation of identical, non-significant stimuli which elicits a decreasing response intensity with progressively smaller reactions). VR is efficient for treating post-traumatic stress disorder [39] and for military training [40]. The affective virtual reality system (AVRS) [41] contains eight VR scenes, rated in terms of valence, arousal, and dominance by 100 subjects.

Although VR is very attractive, there are some limitations, such as the cost of the VR devices, the computing power required by the VR software, motion sickness, size and weight of the head-mounted displays, graphical limitations, potential addiction or eye damage caused by the VR lenses are placed very close to the eyes [42].

## 4. Biophysical Signals

### 4.1. Electrodermal Activity

Electrodermal activity (EDA) or Galvanic Skin Response (GSR) measures the changes in conductance at the skin surface due to eccrine sweat production. EDA is a good biomarker of autonomic system response, being a ”window on the arousal dimension of emotion” [43]. The eccrine glands are the main sweat glands in the body being active in the biological adaptative response, by thermal and emotional sweating. With a total number of approximately 2 to 5 million, the eccrine glands have the highest density in the palm and in the soles, then on the head, but much less on the trunk and in the extremities. Males and females have the same number of sweat glands, the size, and volume of the sweat secreted by each gland, being about five times greater in males [44]. EDA reflects only activity within the sympathetic branch of the Autonomic Nervous System (ANS), being involved in emotional-induced sweating in response to various stimuli [45,46]. Modern functional brain imaging and neurophysiological techniques [43]. have demonstrated that the EDA mechanism is highly modulated within the limbic system via the hypothalamus and the thermoregulatory pathways (for orientation and defensive responses) and to a lesser degree by the premotor cortex and the basal ganglia (involved in the control of limb movements) [47]. Consequently, EDA recordings allow a non-invasive functional assessment of gross movements, attention, emotion perception, threat responses, affective and motivational processes, orientation, decision-making, and fine control [48,49].

EDA continues to be used in psycho-behavioral studies, allowing the assessment of brain activity and autonomic arousal. It is a valuable tool, as it is generally difficult or impossible to influence or control the skin response to emotional stimuli [50]. Figure 1 presents the diagram of the EDA non-stationary signal that decomposes the response to specific tasks into two quantitative measures: *The tonic component*—Skin Conductance Level (SCL) and *the phasic component*—Skin Conductance Responses (SCR), which needs to be evaluated separately [51].

The tonic component is not associated with the onset of a specific stimulus. It is a measure of slow changes in EDA (minutes instead of seconds) and background characteristics of the signal, which specifically refer to the “level” of skin conductance, usually expressed in microSiemens, µS. The changes in the SCL reflect general changes in autonomic arousal, such as emotional state and stress level [52]. The tonic level of skin conductance can vary slowly over time in a person, depending on age, psychological state, hydration, and self-regulation.

The *phasic component* is associated with a specific and identifiable stimulus, being a measure of rapid changes in EDA (measured in seconds) that result from a momentary sympathetic activation. It is referred to as the “response” of skin conductance—SCR [53].

Mean SCL values usually range between 2 and 20 µS. Maximum SCR amplitudes vary within 1 and 3 µS [6]. The Event-Related SCR (ER-SCR) is the sensitive component to emotionally specific events. After a latency period of about 1–3 s (with an average value of 1.8 s in optimal environmental conditions), SCR shows a steep onset, with a typical rise time of about 1 to 3 s, reaching an amplitude of at least 0.1 μS, a flatter recovery and an exponential decay with a recovery time of half of the SCR amplitude between 2 and 10 s, as can be seen in Figure 2. A minimum of 0.05 or 0.04 µS is typically set as a threshold to define a significant SCR [51].

Spontaneous fluctuations in electrodermal responses are known as Non-Specific Skin Conductance Responses (NS-SCRs). They are phasic increases in EDA that have the same appearance as stimulus-elicited EDA responses, but because they are not associated with any specific stimuli or artifacts, they are considered to belong to the tonic component of skin conductance [51]. NS-SCR appears 5 s after the stimulus finished [6]. The typical frequency of NS-SCR is 1–3 per minute during resting periods and over 20 per minute in high arousal situations. An increased frequency of NS-SCRs is considered to be a biomarker of stress, emotions, and anxiety.

Habituation was considered to be a basic way of learning, and SCR was a frequently used indicator for it. [47].

Experimental research has concluded that some people have special electrodermal signal characteristics, similar to a signature, that is influenced by both genetic and environmental factors [54]. A certain category of people has a very small variation of the signal when they are stimulated, compared to the resting state, with fewer NS-SCRs and a more rapid specific EDA habituation. They are classified as “stable individuals”. Opposed, there is the population category with increased reactivity to stimuli, with a higher rate of NS-SCRs and slow habituation to specific stimuli. They are classified as “labile individuals” [55]. There have been attempts to associate these types of electrodermal responses with certain personality traits [47]. People described as calm, deliberative, restrained, good-natured, cooperative, and responsible are the EDA-labiles, while people who are active, emotionally expressive, animated, assertive, more irritable, antagonistic, impulsive, and irresponsible are EDA-stabiles. Some individuals who show electrodermal hypoactivity can have affective disorders with vulnerability for depression or suicide attempts [47,56].

The phobic reaction is a defense-like overreaction to specific stimuli [6]. Several studies have shown that phobic patients do not show a general increase in electrodermal activity, but there is an overreaction only to specific stimuli. The increased amplitude of SCR may be a sensitive indicator for fear responses within the phobia condition. SCL and NS-SCR increased significantly during phobic stimuli presentation [57,58]. The results obtained by Herbelin et al. [24] suggested that SCL and the standard deviation of the heart rate are relevant for assessing emotion intensity. In [31], the percentage change of skin resistance showed significant differences between the nonphobics and phobics subjects after a VR therapy for fear of flying. The phasic component was revealed when arousing stimuli has been presented, although they also occurred spontaneously in some individuals [59,60,61].

### 4.2. Heart Rate

The basic heart rate (HR) is provided by the cardiac conduction system, a network of specialized cardiac muscle cells that ensures the *intrinsic innervation* of the heart and that spontaneously initiate and transmit the electrical impulses responsible for the coordinated contractions of each cardiac cycle. The sinoatrial node, part of the cardiac conduction system (cardiac pacemaker), has its own frequency that varies between 95 and 110 beats per minute, being dependent on age. The ANS ensures the extrinsic innervation of the heart, modifying the activity of the intrinsic system, according to the needs of the organism. The resting heart rate is within the range of 60–70 beats/min, with obvious domination of parasympathetic influences on the sinus node [62]. Sympathetic influences increase the pulsation rate. Assessment of heart rate variability (HRV) is a non-invasive method that actually describes the oscillations between consecutive R-R intervals of the electrocardiogram, as a result of the modulatory effect of the ANS and of catecholamines circulating in blood on cardiac activity. This fluctuation, known as the HRV index, is an indicator of the body’s ability to react to the variations of the external and internal environment [63,64].

The vagus nerve, parasympathetically innervates many of the body’s organs, being responsible for most of the extrinsic innervation of the heart. Distributed especially to the nodules of the cardiac conduction system, the vagus actively attenuates the influences of the sympathetic nervous system (SNS) on the heart. HRV at rest is indicative of ANS health and higher emotional well-being, HR reactivity being a change from the basal state in response to stimuli. When an individual is threatened, the vagal tone is inhibited, and SNS is activated, triggering a suite of responses that promote survival, which affects the decrease of HRV. Polyvagal theory [65] associates higher levels of HRV with the capacity for social engagement, and the neurovisceral integration model (NIM) [66] postulates that higher resting HRV indicates tendencies for appraisal [67]. High HRV, (the parasympathetic influence on heart rate), is associated with a range of physiological processes, adaptive responses, and good adaptability to changes [68]. Related to the emotional state, high HRV is correlated with low anxiety and reduced incidence of phobias, low rumination, and a low level of worry [69]. Low HRV at rest, characterizing the SNS activity on heart rate, tend to be less flexible both physiologically and behaviorally in adapting to environmental demands [70]. Low HRV at rest was associated with difficulties in emotional regulation [71] and psychiatric conditions (depression, anxiety, and alcohol use disorders) [72,73] Very low HRV is conditioned by the humoral influence on HR [62].

### 4.3. Electroencephalography (EEG)

EEG is an electrophysiological method of recording the electrical activity of the brain and is done by attaching electrodes to the scalp, according to a standard format. The various patterns of electrical activity are called waves or brain oscillations and can be identified by amplitude and frequency. Each category of oscillations is associated with certain mental states. For example, the alpha waves (8–13 Hz) are related to states of relaxation, meditation, or mental rest [74]. The EEG measurements are usually based on contrasting the activations of fairly large regions of the brain [75,76]. Initial studies related frontal asymmetry to emotional valence, the left hemisphere is associated with positive emotions, and the right hemisphere with negative emotions. Subjects with higher left frontal activity were reported higher levels of general well-being [77], as they had stronger positive affective reactions to videos with more pleasant content and negative reactions less intense in those with aversive content [78]. Subsequent studies have shown that the method of frontal asymmetry reflects the balance between tendencies of proximity and distance rather than emotional valence [79].

## 5. Protocols for Physiological Data Acquisition in VR

PhysioVR [2] is an open-source framework that facilitates the integration of physiological signals in mobile VR applications. It enables connectivity with the Unity 3D game engine, external event triggering and acquisition, streaming and recording of physiological signals. PhysioVR is composed of two software layers: PhysioSense, which synchronizes the devices and streams data, and PhysioAdapt, which receives and adapts the biophysical signals in the Unity 3D game. By using the Wizard of Oz methodology [80], stimuli can be created, and events can be triggered in real-time. Data flows in real-time between the devices, and the physiological signals can be plotted in real-time along with their corresponding stimuli or events. The protocol has been tested using a VR game called EmoCat Rescue, where the players were required to find a lost cat by listening to stereoscopic sounds. The application offered real-time feedback on the cardiac rhythm, encouraging messages and indications about carrying out breathing exercises that would reduce irregular heartbeats.

Rehabnet Control Panel [81] enables interfacing of physiological sensors with VR applications using the Virtual-Reality Peripheral Network (VRPN) and User Datagram Protocol (UDP). It allows the recording of biophysical data, game events, and visual stimuli.

## 6. Proposed Methodology for Physiological Data Acquisition in VR

The increasing use of VR immersion technology for various applications, especially for detecting anxiety in the biomedical field, requires standardized protocols to address the special steps for data collection and management. The measurement of physiological parameters, objective indicators of the level of anxiety, are unique in VR experiments, because of the particularities of this type of application. We can mention here the need for preparatory steps for: Familiarizing the subject with the equipment, obtaining a baseline to remove interindividual variability, identifying irresponsive people, efficient signal filtering to eliminate power grid noise, etc.

Additional hardware considerations, the need to evaluate anxiety in real-time for the correct guidance of the therapy session, the long duration of the acquisition session, the large volume of resulting data, or the numerous artifacts caused by the subject’s dynamics during the experiment are some of the particular restrictions that occur in this type of experiment.

In this article, we propose a protocol that takes into account the most common of the particular constraints that occur in this type of experiment. At the same time, this paper provides a series of recommendations and suggestions to minimize these shortcomings and to draw up a checklist of standardization procedures for data collection and management, with a focus on assessing anxiety.

Figure 3 presents the diagram of the proposed experimental design and protocol for emotion/anxiety detection in VR.

### 6.1. Preparation of the Experiment

Before carrying out the experiment, a stage of preparation is indicated. This stage has to be scheduled before the experiment, not mandatorily in the same day. Certain aspects must be taken into account:Obtaining informed consent from the participants after explaining the aim and conditions of the experiment helps ensure their well-being and maintains high ethical standards.Filling in a demographic questionnaire specifying age, race, ethnicity, gender, marital status, income, education, employment, health status, previous experience in the virtual environment gives a lot of information about the subjects and allows for the analysis of statistical correlations. In the case of VR applications for emotional evaluation in phobia, anxiety, stress, etc., a short psychometric evaluation is required by applying dedicated questionnaires. They are designed to assess the characteristic symptoms and severity of emotional impairment, using subscales as fear, avoidance, physiological arousal to obtain a total score.A preliminary test or a pilot study, to familiarize the subject with the experimental device (especially in the case of VR equipment) will avoid the stress induced by the impact with the unknown devices. This will help the researchers to find out how long the participants felt comfortable while watching immersive videos before they experienced fatigue or motion sickness [82].It is also necessary to verify and classify individual reactivity, followed by excluding individuals without reaction, because for these individuals, the EDA analysis does not provide too much information. The evaluation of individual reactivity can be done by evaluating the intensity of the physiological responses during the subjects’ exposure to images with known emotional value. About 10% of the participants can be non-responders [52]. This stage can also provide information about the “unstable” or “stable” nature of the subjects’ physiological response.

### 6.2. Data Acquisition

In order to ensure that quality data is acquired both in the stage of preparation, and especially during the experiment, two important conditions should be fulfilled:The design of an appropriate environmental setup, by creating a relaxing atmosphere for the participants that includes a quiet, noise-free environment (with a minimum circulation of people), with an optimal ambient light that ensures emotional balance and reduced unwanted external stimuli. Maintaining the temperature at a constant value (around 23 °C) and the humidity between 40 and 60 ensures that the body functions at the optimum metabolic rate. The period of day (morning, afternoon, or night) should be standardized within the same research project, considering the possible influences of the circadian rhythms [83].The setup of the recording device and the placement of the Ag/AgCl electrodes should be done correctly. There are several configurations for their positioning, which involves the regions where the eccrine sweat glands have a higher density: The fingers, palms, or soles. In VR applications, mostly finger electrode placement is used. The sensors are placed on the index and ring finger for better cable management. It is recommended to place the electrodes on the adjacent fingers of the non-dominant hand (second and third fingers, or fourth and fifth fingers), because they are innervated by the same spinal nerve (median respectively ulnar) [84]. Due to the greater number of sweat glands, it is recommended to fix the electrodes on the distal phalanx of the fingers. Another important aspect at this stage is to avoid data recording errors resulting from the improper connection of the electrodes (either by faulty locations or by the lack of contact with the skin) and to remove possible electromagnetic interference. This requires cleaning the skin before placing the electrodes (it is recommended to clean only with water), using high-quality electrodes, fixing them very well, and in some situations, even using electroconductive gel in contact with the skin. Before beginning the recording sessions, it is necessary to stabilize the skin-electrodes interface. The researcher has to *check the sensors’ sensitivity*, by asking the participant to take a deep breath in and to hold it for a few seconds. A good EDA signal should contain a spike in skin conductance within 2–3 s after the breath was initiated [51].

During exposure therapy, the physiological parameters can be monitored by techniques such as EDA, HR, EEG, etc. For the daily practice of a psychotherapist, the use of EEG is more difficult, the installation of electrodes being time-consuming and introducing an additional source of stress. The VR device can be an obstacle to the correct placement and stability of the electrodes. Therefore, in the proposed protocol, we will focus mainly on EDA and HR.

Baseline recording. At the beginning of the experiment, once the physiological sensors and VR equipment are placed, it is advised that the subject should wait several minutes (typically 5–15 min) before beginning the recording session, in order to stabilize the skin-electrode interface [53]. Once the interface was stabilized, to obtain correct baseline values, the participants should be asked to relax for 60 s, without ample movements, without breathing deep, while watching neutral images. In this way, the non-specific signals are reduced. At the end of the baseline recording, it is indicated to apply a stimulus to obtain a reference response—for example, during a deep breath.VR exposure represents the actual section of therapy, which consists of the active participation of the subject in a scenario that involves the confrontation with phobic stimuli. The sensors acquire biophysical data, and analysis should be performed to highlight the relation between occurring events and the measured data. To maximize the therapeutic effect, the scenario should be adapted to the subject’s condition, assessed through the degree of anxiety. The adaptation of the scenario aims to gradually exposure of the subject to the stimuli, thus avoiding unwanted situations of overreactions (panic attacks). Therefore, the real-time assessment of anxiety levels is mandatory.

### 6.3. Data Processing

#### 6.3.1. Data Preprocessing

Raw biophysical data is preprocessed by following the next steps:

Data filtering refers to removing unwanted components from a signal. The filter is used to smooth noisy signals and signals unrelated to a specific analysis [85]. There are more filters that can be used in data processing: Basic median filters, low-pass filtering, exponential smoothing, Kalman filter [86].By analyzing the EDA signals acquired during various experiments, we found that the noise with the highest amplitude is the one induced by the electrical network. The main component of this type of noise has a frequency of 50 Hz, but it may also contain significant harmonic components generated by non-sinusoidal currents of nonlinear loads. The high frequency of these components, relative to the spectrum of the EDA signal, allows their efficient filtering by using a low-pass filter. However, it is very important to use a proper sampling frequency in order to avoid the aliasing effect. According to the Nyquist–Shannon sampling theorem, the minimum sampling frequency must be 100 Hz, to avoid aliasing at the fundamental frequency, but it is advisable to choose a higher frequency due to the presence of the harmonic components.The first step in implementing the low-pass filter is to decide between the two main categories of filters: Infinite Impulse Response (IIR) and Finite Impulse Response (FIR) filters. Although the IIR filters are much more efficient in terms of computational effort, required memory, and have a shorter response time, we recommend using FIR filters because they have a shorter transition band, are much more stable numerically, and introduce fewer shape distortions of the signals due to their linear phase characteristic.The heart rate information taken from the data acquisition devices is the result of complex processing (that includes filtering), performed on electrocardiogram (ECG) or photoplethysmographic (PPG) signals. For this reason, additional filtering of the HR signal is often not necessary.The noise in the EEG signals is issued by eye blinks and body movements. In addition to the recommendation to minimize participants’ movements, more techniques, such as least mean square and blind source separation, are used. Moreover, filter techniques are applied: 4 Hz, 5Hz, or even lower frequencies such as 0.5 Hz, 1 Hz as the lower-band cut-off frequencies; 30 Hz, 70 Hz as the upper-band cut-off frequencies. For the database construction, a 5th order Butterworth band-pass filter, 4 Hz lower-band, and 45 Hz upper-band filters are applied [87].The downsampling of a data sequence is the process of reducing the sampling rate [50]. This process can be achieved in multiple ways. The simplest of them is the sampling rate reduction by an integer factor, known as the “decimation” method. Decimation with an integer factor n consists of taking a sample from a data sequence and discarding n − 1 samples for every n samples [50].Downsampling is usually required when the signal’s sampling rate is higher than the one imposed by the bandwidth of the signal. In the case of the EDA signal, the efficient elimination of the noise generated by the electrical network requires the use of such a high sampling frequency. Because further processing of the EDA signal requires a computational effort that depends heavily on the number of samples, downsampling is a necessary condition for ensuring the real-time processing of this signal. In order to avoid the aliasing effects caused by downsampling, it is necessary to reduce the sampling rate down to at least double the low-pass filter frequency limit.Normalizations and standardization refer to data transformation to perform good statistical data analysis. Normalizations means rescaling the values to fit in a range, including data transformation to correct for skew/kurtosis so that the data is fit for parametric statistical analysis (min-max, Z-score, log transformation, or square-root transformation).Standardization refers to another data scaling technique, to perform directly and meaningfully a comparison between subjects. Standardization may not always be necessary, but the most common data transformations include: Transformations into standard values (min-max, Z-score), Range-Corrected Scores, Proportion of Maximal Response, etc. [52,87].

#### 6.3.2. Data Analysis

Deconvolution. Skin conductance data is characterized by the overlap of two components: A slowly, continuously varying tonic component (skin conductance level—SCL) over which a rapid component overlaps (phasic skin conductance response—SCR), in close correspondence with the arousal states, whose amplitudes give information on the intensity of the emotional states. While data acquisition can be made quite easily, data analysis is a complex process, through which the discrete reactions superimposed over the tonic component must be decomposed. The variability of SCR shapes hereby complicates the proper decomposition of SC data. Benedek [88,89] proposes a method for the full decomposition of the SC data into tonic and phasic components, starting from the biological model of sweat diffusion, a model with two compartments. Two EDA analysis strategies are proposed: Continuous Decomposition Analysis (CDA) and Discrete Decomposition Analysis (DDA). CDA [88] is the recommended method for the analysis of skin conductance data. This method extracts the phasic information underlying EDA, deconvolving the SC data by the general response shape. The results of a large increase of temporal precision, the data is decomposed into continuous phasic and tonic components, with several standard measures of phasic EDA. DDA [89] uses non-negative deconvolution to decompose SC data. It is an algorithm difficult to use for real-time estimates, because it may be slow for large data samples.Features extraction is an essential step for emotion classification. Starting from an initial data set, after processing the signals, a multitude of derived data, called features, can be extracted and used to estimate the anxiety level. Before extracting the features, a segmentation stage of the collected data is recommended, by extracting and analyzing parts of the raw physiological signal, called time-windows. The length of the window can be different: The same as the stimuli length, shorter or longer than stimuli length. Not all the features have the same relevance for estimating anxiety, and previous works have made limited contributions to identifying the most appropriate ones. In our study, we focus only on the features extracted from the skin conductance and heart rate related data, and we will try to identify the most relevant ones for assessing the level of anxiety in real-time.

### 6.4. Classification

The last step in this protocol involves the use of the features extracted from the measured physiological parameters in order to obtain an estimate of the anxiety level. Subsequent use of this level of anxiety to adapt to the scenario of VR exposure therapy can be facilitated by classifying it into several intensity categories according to an evaluation scale. Among the algorithms that obtained a high classification accuracy are the classical machine learning methods and the convolutional neural networks (CNNs) [90,91].

## 7. Case Study for Acrophobia Therapy

The aim of the case study consists, first of all, in validating the data acquisition protocol presented above. Moreover, we want to determine the physiological parameters that have the highest degree of correlation with the level of anxiety, which will be further used for estimating in real-time the subject’s emotional states.

### 7.1. Description of the Experiment

An experiment was performed to expose the subjects to stimuli that can trigger the fear of heights (acrophobia), the exposure being achieved by immersion in VR.

A group of seven subjects, aged 22–50 years, was selected for the experiment. After completing the Visual Height Intolerance questionnaire [92] that assessed the degree of anxiety during heights exposure, the subjects have been classified into three categories: Two suffered from a mild form of acrophobia, three from a medium-intensity fear of heights, and two experienced a severe form of height intolerance. More details can be found in [93,94,95]. To familiarize the subjects with the experimental device and with the VR perception, we provided some VR introductory sessions. Before starting the experiment, all the subjects were informed about the experiment and signed informed consent. The experiment was approved by the ethics committee of the UEFISCDI project 1/2018 and UPB CRC Research Grant 2017, and University POLITEHNICA of Bucharest, Faculty of Automatic Control and Computers.

The experiment was designed as a game with visual and vestibular stimuli, rendered on the HTC Vive head-mounted display [96], in which the subjects had to collect 15 coins with different values (bronze, silver, and gold), situated on the ground floor and on the balconies of a building (at the first, fourth and sixth floor). Each time the user collects one coin, a virtual panel appears and asks the participant to rate the level of fear perceived on a gradual 11-choice scale called Subjective Unit of Distress (SUD) (0 for complete relaxation, 1–3 for low fear, 4–7 for medium stress level and 8–10 for high anxiety) (Figure 4).

In order to evaluate the reliability and internal consistency of the self-reported anxiety levels, we performed a Cronbach alpha test (using IBM SPSS^®^ Statistics software version 27.0). The resulted alpha coefficient was 0.91, which suggests that the self-reported data to have relatively high internal consistency.

Throughout the whole game, peripheral physiological data were recorded via a Shimmers Multi-Sensory device [97]. The Shimmer3 GSR+ Unit measures the skin conductance in microSiemens, GSR data being acquired at a rate of 51.2 samples per second and uses an optical pulse probe to capture a PPG signal that is later converted to estimate heart rate (HR). The data are sent via Bluetooth connectivity to a computer. Each time a coin is collected in the game, a timestamp is added to the file in which the recorded physiological data is saved.

During some experiments performed at the beginning of this study, we found the presence in the EDA signals of some periodic disturbances with amplitude and a period close to that of the electrodermal responses. Figure 5 shows a segment of the noise-affected EDA signal corresponding to a baseline recording.

The noise signal is generated by the electrical network, although, at first sight, the fundamental frequency of this signal (1.168 Hz) would exclude this cause. In reality, the recorded signal is the result of the aliasing effect introduced by incorrect signal sampling.

Because the sampling frequency used to record the signal was 51.2 Hz, the alias mechanism, shown in Figure 6, explains the presence of components with frequencies of 1.168 Hz, 2.337 Hz, …, in the spectrum of the acquired signal as aliases of a fundamental frequency of 50.032 Hz and its harmonics.

In order to avoid this phenomenon, we recommended in the protocol the use of a sampling frequency much higher than twice the frequency of the noise generated by the electrical network, low-pass filtering to eliminate this noise, and downsampling of the filtered signal to return to a lower sampling rate compatible with the bandwidth of the EDA signal.

The data collected was processed to estimate the degree of anxiety reached by the participants during the game. At the same time, the predictive accuracy of the proposed protocol was evaluated.

### 7.2. Data Analysis

Data analysis has consisted of several steps: The deconvolution of the GSR signal, data segmentation, features extraction, the selection of the most informative features, and the determination of a correlation between the extracted features and the anxiety levels.

#### 7.2.1. Deconvolution

The deconvolution of the GSR signal has been processed using the Continuous Decomposition Analysis algorithm [88] implemented in Ledalab V3.4.9 [98]. This algorithm extracts the tonic (SCL) and phasic (SCR) signal components and also identifies the amplitude (AMP) and onset time (TON) of electrodermal responses.

#### 7.2.2. Data Segmentation

A data window of 5 s before the moment when the subject picked up a coin was extracted from all signals. This window corresponds to the maximum exposure to the phobic stimulus. For each time segment, we had a self-evaluation rating of the anxiety level between 0 and 10, because the subject was asked to rate his/her anxiety level after picking up each coin.

#### 7.2.3. Features Extraction

As we have seen from the literature presented, a wide range of features can be extracted from raw data. But not all the parameters that can be extracted have the same importance in interpreting the biological significance of the data. Therefore, identifying the most relevant features or finding an optimal set of features that provide fast and accurate information on estimating anxiety is of major importance, especially when processing experimental data in real-time.

Thus, in our experiment, starting from the raw data, an extended set of features has been extracted. In this stage of the study, we selected all the features, which were suggested more frequently in the literature. In the next stage, following the feature extraction, an exhaustive search will be performed to select the most relevant features for anxiety level estimation.

The extended set of features contains three categories of features that characterize the biophysical signals in terms of time evolution, GSR response events, and frequency content as following:

##### Electrodermal Activity—Time Domain Features

The time domain features represent a series of parameters that are easily calculated, being descriptors of the waveform amplitude, frequency, and duration [99]. The signals are defined by their samples: ${\left\{x_{K}\right\}}_{K=0\ldots N-1}$, where *X_k_* is the *k*-th sample of the signal *X*, and *N* represents the number of signal samples inside the time segment.

(1)
*MAV— mean absolute value*
(1)$$X\_MAV=\frac{1}{N\;}\sum_{K=0}^{N-1}\left|x_{K}\right|,$$
(2)
*MAR—mean absolute value relative to baseline*
(2)$$X\_MAR=\frac{1}{N\;X_{B}}\sum_{K=0}^{N-1}\left|x_{K}\right|,$$
(3)*STD—standard deviation*(3)$$X\_STD=\sqrt{\frac{1}{N-1}\sum_{K=0}^{N-1}{\left(x_{K}-\bar{x}\right)}^{2}},$$
where $\bar{x}=\frac{1}{N}\sum_{K=0}^{N-1}x_{K}$ represent the mean value of the signal.(4)*WL—waveform length* [100]
(4)$$X\_WL=\sum_{K=1}^{N-1}{\left(\Delta x_{K}\right)}^{2},$$
where $\Delta x_{K}=x_{K}-x_{K-1}$.(5)*SSC—slope sign changes*(5)$$X\_SSC=\sum_{K=1}^{N-2}f\left(x_{K-1},x_{K},x_{K+1}\right),$$
where $f\left(x_{K-1},x_{K},x_{K+1}\right)=\left\{\begin{array}{l} \begin{array}{c} 1\;if\;sign\;\left(x_{K}-x_{K-1}\right)\cdot sign\;\left(x_{K+1}-x_{K}\right)<0\;and\\ \left|x_{K}-x_{K-1}\right|\geq \varepsilon \;and\\ \left|x_{K+1}-x_{K}\right|\geq \varepsilon \end{array}\\ 0\;otherwise \end{array}\right.$ and ε=0.001 μS.(6)*WAMP—Willison amplitude*(6)$$X\_SSC=\sum_{K=1}^{N-1}f\left(x_{K-1},x_{K}\right),$$
where $f\left(x_{K-1},x_{K}\right)=\left\{\begin{array}{l} 1\;if\;\left|x_{K}-x_{K-1}\right|\geq \varepsilon_{W}\\ 0\;otherwise \end{array}\right.$ and $\varepsilon_{W}=0.5\;\mu S$.

##### Electrodermal Activity—Events Related Features

(1)*NR—number of responses—*represents the number of electrodermal responses detected inside the analyzed time segment
(7)$$X\_NR=length\;\left(AMP\right)=length\;\left(TON\right),$$(2)
*AVRA— average response amplitude*
(8)$$AVRA=\frac{1}{NR}\sum_{K=0}^{NR-1}AMP_{K},$$
(3)
*MAXRA—maximum response amplitude*
(9)$$MAXRA=\underset{K=0\ldots NR-1}{\max\;}\left(AMP_{K}\right),$$


##### Electrodermal Activity—Frequency Domain Features

The calculation of these parameters is a more laborious process because it requires evaluating the power spectrum density (PSD) of the electrodermal signals (GSR, SCL and SCR). For each data window, PSD is evaluated for a set of frequencies ${\left\{f_{K}\right\}}_{K=0\ldots N-1}$ between 0 and 25.6 Hz with a 0.05 Hz step.

(1)*FMD—frequency median* [99] represents the frequency that divides the signal spectrum in two sections that have the same energy and is located in the frequency set used for the PSD calculation at the index *M* which minimizes the value:(10)$$\left|\;\sum_{K=0}^{M-1}P_{K}-\sum_{K=M}^{N-1}P_{K}\right|,$$(2)*FMN—frequency mean* represents the average frequency weighted by the power spectrum density:(11)$$f_{MN}=\frac{\sum_{K=0}^{N-1}f_{K}P_{K}}{\sum_{K=0}^{N-1}P_{K}},$$

##### Heart Rate—Time Domain Features

Heart rate is defined as the time interval between successive heartbeats. For each data, window corresponds a vector ${\left\{h_{K}\right\}}_{K=0\ldots P-1}$ that contains the heart rate values.

(1)
*MAV— mean absolute value*
(12)$$HR\_MAV=\frac{1}{N}\sum_{K=0}^{P-1}\left|h_{K}\right|,$$
(2)*STD—standard deviation* is a measure of heart rate variability and is defined as:(13)$$HR\_STD=\sqrt{\frac{1}{N-1}\sum_{K=0}^{P-1}{\left(h_{K}-\bar{h}\right)}^{2}},$$
where $\bar{h}=\frac{1}{N}\sum_{K=0}^{P-1}k_{K}$ represent the mean value of the signal.

##### Heart Rate—Frequency Domain Features

In order to obtain this class of features, the heart rate signal ${\left\{h_{K}\right\}}_{K=0\ldots P-1}$ is decomposed in two signals ${\left\{hl_{K}\right\}}_{K=0\ldots P-1}$ and ${\left\{hh_{K}\right\}}_{K=0\ldots P-1}$ that contain the frequency components between 0.04 Hz and 0.15 Hz, respectively, between 0.15 Hz and 0.4 Hz [101]. The signal decomposition is performed using two 3rd order band-pass Butterworth filters, as we can see in Figure 7.

(1)*LFI—low frequency signal intensity* represents the intensity of the signal containing frequency components between 0.04 Hz and 0.15 Hz:(14)$$HR\_LFI=\sqrt{\frac{1}{P}\sum_{K=0}^{P-1}hl_{K}^{2}},$$(2)*HFI—high frequency signal intensity* represents the intensity of the signal containing frequency components between 0.15 Hz and 0.4 Hz:(15)$$HR\_HFI=\sqrt{\frac{1}{P}\sum_{K=0}^{P-1}hh_{K}^{2}},$$(3)*HLR—high/low frequency intensity ratio* represents the ratio between intensities of low frequency and high frequency HR signals:(16)$$HR\_HLR=\frac{HR\_HFI}{HR\_LFI},$$

For each data window of 5 s, the following features were extracted from the electrodermal and heart rate signals as presented in Table 1.

As shown in Table 2, for each subject, we obtained 15 time windows, with 32 numerical values of the features for each window. Each window has an associated level of anxiety (ANX), resulting from the self-assessment. Consequently, there are 105 data points consisting of the feature values and the corresponding self-reported anxiety levels.

#### 7.2.4. Selection of the Most Informative Features and Finding the Relation for Anxiety Estimation

A prospective study to identify the features which are useful in estimating anxiety levels began with a Pearson bivariate correlation between anxiety and feature. Table 3 presents a ranking with the features that showed the highest degree of correlation with the subjects’ self-expressed anxiety levels.

It can be observed that more than half of the features present a high degree of correlation with the level of anxiety. This suggests the possibility of obtaining an estimate of the anxiety level based on the EDA/HRV features values.

There are two possible approaches for the anxiety estimator design:Using a neural network trained with the experimental data;Using a mathematical formula identified based on the experimental data.

The first approach is very commonly used today, but requires a much larger volume of training data than is available in this case study, and the computational effort required for the neural network classification can be an issue for the real-time implementation. Therefore, we decided to use the second approach that involves a selection of a set of relevant EDA/HRV features, the proposal of an estimation formula, and the identification of its coefficients.

Our proposed approach involves a regression analysis for the estimation of the relationship between the anxiety intensity (dependent variable) and physiological signal features (independent variables).

The most important step in this analysis is the choice of the regression model. Most regression models are described by the following equation:(17a)$$y=f\left(x,c\right)+e,$$
where *y* represents the dependent variable, *x*—the vector containing independent variables, *c*—the vector of model parameters, and *e* represents an additive error term that may stand in for un-modeled determinants of y or random statistical noise.

In practice, there are specific models known to be suitable for different categories of processes. In our case, the selection of the regression model is difficult to be done because there is no a priori information about the composition of the features set (which are the features that best predict the level of anxiety) and the actual formula of the function $f\left(x,c\right)$.

Due to unlimited possibilities to create the model function, we chose to use a second order approximation of the multivariable Taylor series:(17b)$$y=f(a)+{\left(x-a\right)}^{T}\nabla f(xa)+{\left(x-a\right)}^{T}H(x_{0})\;(x-a),$$
where ∇f(a) and H(a) are the gradient vector, respectively, Hessian matrix of the estimation function f(x) evaluated in the reference point a.

If the reference point is chosen a=0, due to the symmetry of the Hessian matrix, Equation (17b) can be rewritten as:(17c)$$y=\sum_{\begin{array}{l} I=1\ldots P\\ J=1\ldots P\\ I\leq J \end{array}}\alpha_{I,J}\;x_{I}\;x_{J}+\sum_{I=1}^{P}\beta_{I}\;x_{I}+\gamma ,$$

Coefficients vector $c=\left\{\;\alpha_{I,J},\;\beta_{I},\;\gamma \right\}$ can be determined using the least squares ethod to minimize the sum of squared residuals:(17d)$$S=\sum_{K=1}^{N}\;{\left[\;y(k)-y^{*}(k)\;\right]}^{\;2},$$
where y(k) and $y^{*}(k)$ are the real values of the dependent variable from the available dataset, respectively, the model output values. In our case, because the dependent variable is the anxiety level y(k)=ANX(k) and $y^{*}(k)=ANX^{*}(k)$ represents the self-reported, respectively, estimated anxiety levels corresponding to the data point from line k in Table 2. Moreover, the regression model function acts as an estimation for anxiety level based on the values of biophysical signals features.

The most difficult part is the selection of the features that compose the estimation function. Using all features would be difficult, due to the high complexity of the estimation function (if all 17 significant features are used, the estimation function would have 171 coefficients).

In order to select the optimal combination of features, we performed an exhaustive search using a MATLAB script specially developed for this purpose. The algorithm used is the following:

Randomly split the experimental dataset into two subsets: Identification set (IS) containing 90 data points and validation set (VS) containing 15 data points;For all combinations of P features/from the set of 32 (there are/possible combinations):
Determine the coefficients of the anxiety estimation function using the current feature set and the information from the IS data points;Evaluate the sum of squared residuals;Store the current features names, estimation function coefficients/and the sum of squared residuals S;Find the best selection of features and the corresponding evaluation function coefficients by searching the minimum sum of the squared residuals in the stored data;Using the optimal set of features and the corresponding estimation function, evaluate the sum of the squared residuals for the VS data points in order to validate the efficiency of anxiety estimation.

The script was run with a different number of features, starting from P=1 up to P=7. For a large number of features, the process of selection became very time-consuming (if P=8, there are more than 10 million feature combinations to analyze).

A very important part of this algorithm is the splitting of the dataset into the identification set (IS) and validation set (VS). If the identification of these coefficients had been made to minimize the sum of squared residuals for the entire set of experimental data, there would have been a risk of overfitting. Thus, the estimation function would have presented very good performances for the data in this set, but the accuracy of the estimation would vary widely in the case of the data sets generated during new experiments.

The solution used to reduce this phenomenon is to identify the function coefficients using only a part of the data set and validate the estimation accuracy to be done using the rest of the data. The success of this approach depends on how the data set is separated, so that both subsets are representative. If this condition is met, the degree of accuracy obtained for the data in the identification set and those in the validation set is similar.

The attempt to randomly separate the experimental data set obtained in the case study frequently led to excellent results for estimating anxiety on the training set, but low accuracy for the validation set (an example is shown in Figure 8).

The occurrence frequency of these situations, as well as the level of accuracy difference between the estimates on the two subsets of data, increases with the increase in the number of parameters used and consequently in the number of coefficients in the estimation function. One explanation for this phenomenon would be that a more complex function is much more likely to overfit to a small set of identification data. For this reason, the use of a larger number of parameters is not indicated if a large number of experimental data is not available.

To reduce the risk of choosing unrepresentative data subsets, we introduced a condition for their generation, namely, a distribution of reported anxiety levels relatively identical at the level of the two data subsets.

The results presented in Figure 8 correspond to the following distribution of anxiety levels in the two subsets (Table 4).

## 8. Results

In the process of finding the best combination of features to be used in the anxiety estimation function, we performed an exhaustive search over all possible combinations of features. The number of features taken into account as input variables into the estimation function was initially selected to 1 (resulting $C_{32}^{1}=32$ possible choices) and gradually increased up to 7 (resulting $C_{32}^{7}=3365856$ possible combination choices). For each number of input variables, the optimal combination of features and corresponding estimation function coefficients were selected based on the minimization of the sum of squared differences between the reported and estimated anxiety level.

In order to obtain reliable results, the experimental data set was randomly divided into two subsets. The first subset (IS), containing 90 data points, was used for the identification of estimating function coefficients. The second subset (VS), containing 15 data points, was used to assess the accuracy of the anxiety estimation on a different data set than the one used to determine the estimation function.

The accuracy of the estimation was also assessed by means of the sum of the squared differences between the reported and estimated anxiety level. This criterion was evaluated separately for the data in the identification set (IS), validation set (VS), as well as for the entire data set (IS + VS).

Table 5 presents the best accuracy obtained by the optimal estimation function for a different number of features. Moreover, this table presents the names of the features that give that accuracy level.

By analyzing the data in the table, we can observe the small difference between the values evaluated for the set of identification data and the validation one. This observation confirms that each identified function is able to provide estimates with a similar degree of accuracy, not only for the measurements performed in this case, but also in the case of other measurements that will be performed during future VR experiments.

Moreover, as expected, the accuracy of the estimate increases with the increasing number of parameters used. Because evaluating the estimation function does not involve a high computational effort, the function with the highest number of parameters can be used for maximum estimation accuracy.

Another performance indicator is the accuracy of anxiety classification. It can be evaluated by the confusion matrix and the overall classification accuracy (percentage of correct classified anxiety levels).

In our study, we tested anxiety classifications, levels 2 and 3. The intensity intervals definition of the classes are:[0 … 5] for “low” and (5 … 10] for “high”[0 … 3] for “low”, (3 … 7] for “mild” and (7 … 10] for “high”.

The results of estimation evaluation for different numbers of features selected as input values are presented in Table 6, Table 7, Table 8, Table 9, Table 10, Table 11 and Table 12. Each table presents the histogram associated with the distribution of estimation errors (on the scale of 0 to10), confusion matrix for levels 2 and 3, and corresponding overall classification accuracy.

## 9. Discussion

The protocol proposed in this article was developed following the data analysis of the experiments performed in order to optimize the accuracy of anxiety level estimation during VR based exposure therapy. The elements of this protocol were designed in order to allow estimations in real-time with low latencies. Therefore, although the proposed protocol largely follows the structure of most protocols present in the literature (data acquisition, feature extraction, estimation/classification of anxiety level), the need to make real-time estimates led to the use of a shorter time window for feature extraction and a regression model for estimating the level of anxiety (as an alternative to the computationally intensive methods currently used for classification).

The performance evaluation of the proposed anxiety estimation protocol can be made by analyzing two categories of parameters: Estimation error characteristics and classification accuracy.

Analyzing the estimation error, the standard deviation value of 1.02 obtained in the case of an estimation function with seven features makes us to expect that 95% of the estimates do not differ by more than two units from the real values that can be considered a good performance.

If only two classes are used, the classification accuracy evaluated by the confusion matrices is very good even for regression models that use a low number of features (with only three features, the “low” anxiety level was correctly classified in 99% of cases and the “high” level in 75%).

Classification in three classes proves to be a more difficult task, a good accuracy level being reached for a regression model that uses six features (95% correct classification for “low” anxiety level, 76% for “mild” and 100% for “high). Moreover, it can be observed that most of the misclassified values belong to the “mild” class.

Although increasing the number of features has the effect of reducing the estimation error, it can be observed that this improvement does not always reflect in classification accuracy. For example, increasing the number of features from 6 to 7 reduces the standard deviation of the estimation errors from 1.14 to 1.02, but the overall classification accuracy decrease from 92.38% to 90.48%. This phenomenon can be explained by small errors in estimating the level of anxiety can lead to incorrect classifications if its value is located around the threshold.

In the case mentioned above, when seven features were used for estimation, 3 of the 17 levels of anxiety in the "mild" class were incorrectly classified as "low" or "high" as follows: One anxiety level was misclassified as “low” because its estimated value was 2.91 (slight below the threshold value of 3) and two anxiety levels were misclassified as “high” because their estimated values were 7.12, respectively, 7.37 (slight above the threshold value of 7).

Another important observation related to the classification performance is the high level of the overall accuracy obtained even if the correct classification rate is relatively low, especially in the case of “mild” anxiety levels. The cause of this overestimation of accuracy is the unbalanced distribution of anxiety levels in the data set. The prevalence of “low” anxiety levels is very high (78 out of 105) and the accuracy of classifications in these cases is very good (around 95%) that reduces the impact of the few misclassified levels from the other classes.

### 9.1. Comparison with Other Studies Results

Anxiety disorders affect about one-third of the population during their lifetime. For this reason, researchers make efforts in order to identify all possible causes, improve diagnostic accuracy, and find more efficient treatment protocols. Because evaluating the anxiety level is very important, especially during treatment sessions, multiple studies have been done in this area. Most of the recent studies are focused on the use of various physiological signals in order to evaluate anxiety intensity.

Therefore, an objective evaluation of the results obtained in our study requires a comparison with several works. We selected articles that present different anxiety classification methods tested with both real and VR-induced stimuli. Table 13 presents a summary of the comparison between different classifiers found in the literature and our results.

The first observation related to this comparison is that almost all studies use machine learning approaches for classifier implementation. Our approach is different because we chose to estimate anxiety level based on a regression model and after that to decide an affiliation to different classes by simple comparisons with threshold values.

Another observation is related to the length of the time window used for feature extraction and classification. In most of the studies, the time window is relatively large (over two min). A longer time window increases evaluation precision for the frequency-related features, especially if the frequency ranges involved are in the sub-hertz domain. Moreover, large time windows can reduce the effect of the artifacts that may occur during signal acquisition.

In our study, we aimed to perform the anxiety level estimation in real-time because this estimation will be used as feedback to adapt the VR exposure scenario during treatment sessions. In order to obtain a fast response of the estimates to the changes in patient anxiety state, it is necessary to use time windows as small as possible, which led us to choose a time window of 5 s.

Comparative evaluation of the performances obtained in our study based only on the overall accuracy of classification is not very relevant. The overestimation of classification accuracy (94.29% for two classes and 92.38% for three classes) due to the unbalanced distribution of the anxiety levels in our data set requires a more detailed comparison of performance based on confusion matrices.

The best accuracy level was obtained with a regression model that have six features as input variables. According to confusion matrices, the lowest rates of correct classification are:81% for the “high” class—for level 2 classification;76% for the “mild” class—for level 3 classification.

It can be observed that these values are close to the average accuracy levels obtained in the studies presented for comparison in Table 13, therefore we can conclude that our approach is suitable for anxiety level detection and classification.

### 9.2. Limitations

The most important limitation in our study came from the reduced number of subjects involved in the case study. Moreover, only two subjects experienced a severe form of height intolerance that led to the unbalanced distribution of the anxiety levels in our dataset. Therefore, the inclusion of more subjects with different forms of height intolerance is required in order to obtain more reliable results.

This study did not address the effects of possible artifacts induced by the movements of the subjects. The most important source of movement artifacts is the handling of the VR device controllers that can affect the quality of GSR electrodes contact with skin.

Furthermore, in the case of the GSR signal, the extraction of the physiological parameters involved the decomposition and analysis of the signal based on the Continuous Decomposition Analysis (CDA). This algorithm has a high degree of complexity as well a non-causal character (the determination of the current samples of tonic and phasic components requires knowing several future samples from the analyzed EDA signal) that limits its use in real-time anxiety classification.

In this context, we notice the need to develop an algorithm (possibly suboptimal) for the decomposition and analysis of the EDA signal in real-time. The task of developing this algorithm is simplified by the procedure for selecting the optimal parameters for estimating the level of anxiety indicated the need to use only parameters derived from the decomposition into tonic and phasic components and not those derived from events identification.

## 10. Conclusions

Virtual reality is a viable alternative to in-vivo exposure for treating phobias. The possibility of adapting the scenario during exposure therapy, offered by virtual reality, is an important element for increasing the effectiveness of the therapy. The scenario’s adaptation involves modulating the intensity of the phobic stimuli according to the level of anxiety induced by them. For this reason, an objective and real-time assessment of anxiety level based on the analysis of measurable physiological parameters is an essential condition for implementing this type of therapy.

In this article, we have presented a protocol for evaluating the level of anxiety, to be used in phobia therapy with exposure in virtual reality. This protocol contains recommendations on the preparation and measurement of physiological parameters during the therapy session and proposes a method of processing the acquired data in order to estimate the level of anxiety.

Both the recommendations and the estimation method were elaborated following the analysis of the experimental data obtained during the exposure of seven subjects to stimuli that trigger fear of heights.

The method for estimating the level of anxiety is based on the real-time evaluation of a nonlinear function having as parameters a set of features extracted from the physiological signals measured during the therapy session.

The larger volume of data obtained from a higher number of subjects will allow us to increase the accuracy of the estimates and will provide a higher degree of certainty for protocol validation. Moreover, a larger volume of data will allow us to train a neural network as an alternative to estimate the level of anxiety, and then perform a comparative analysis of the results in terms of accuracy and implementation complexity.

The integration of protocol at biophysical measurements in Virtual Reality environments for anxiety detection can have numerous implications in both practice and research.

For clinical applicability, we can mention the treatment of anxiety by the VR immersion therapy, ideal being the development of a system with the dual operation: (i) For subclinical forms, the system can be implemented autonomously, like a biofeedback application or possibly with remote targeting by the therapist; (ii) for the clinical forms, the direct intervention of the therapist is needed, with a control panel for the permanent monitoring of the evolution of the treatment and of the patient’s condition (by monitoring the physiological parameters).

As a research target, the protocol can be used as a method of developing and testing applications that require real-time evaluation of the user’s emotional response (medical, educational applications, military, or sports training).

## Figures and Tables

**Figure 1 sensors-20-07088-f001:**
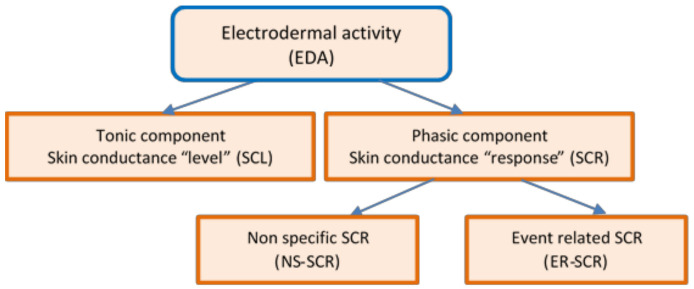
Diagram of the Electrodermal activity signal decomposition.

**Figure 2 sensors-20-07088-f002:**
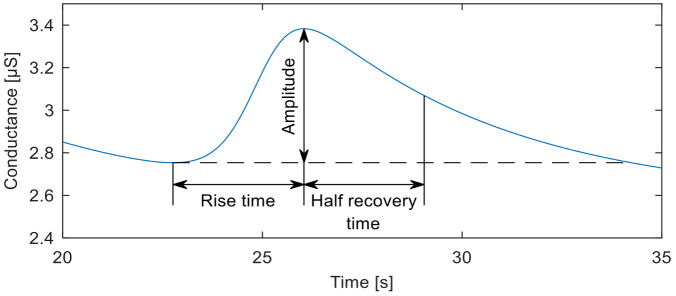
Event-related skin conductance responses SCR.

**Figure 3 sensors-20-07088-f003:**
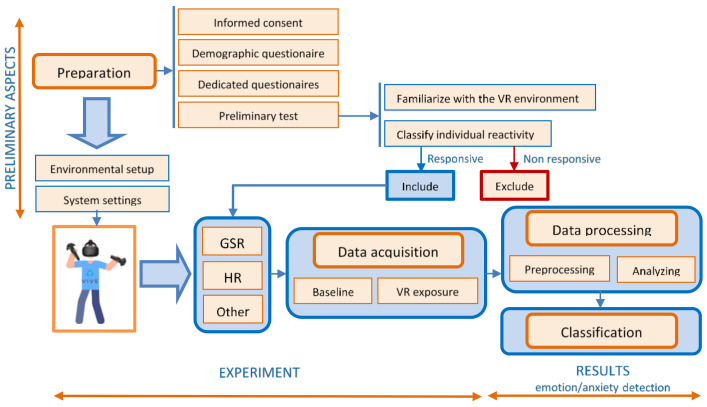
Block diagram of the experimental design and protocol for emotion/anxiety detection in virtual reality (VR).

**Figure 4 sensors-20-07088-f004:**
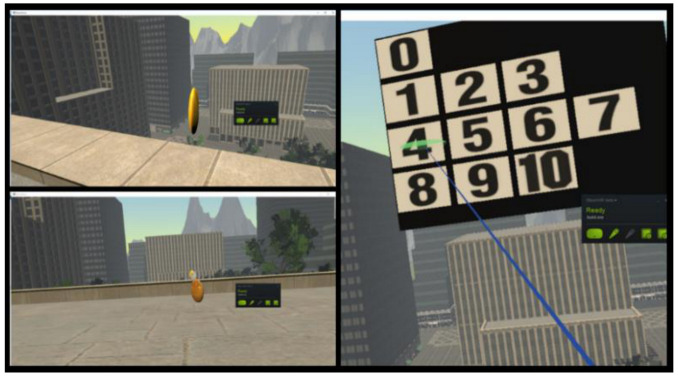
The VR acrophobia game.

**Figure 5 sensors-20-07088-f005:**
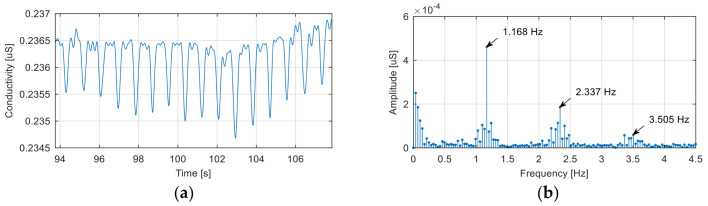
Example of a signal affected by electrical network noise (**a**) signal in the time domain; (**b**) frequency spectrum of the signal.

**Figure 6 sensors-20-07088-f006:**
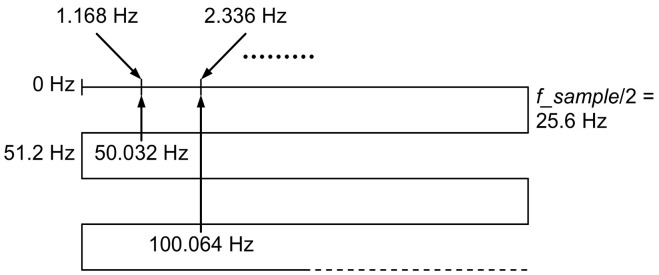
Alias effect during sampling of the electrical network noise with a sampling frequency of 51.2 Hz.

**Figure 7 sensors-20-07088-f007:**
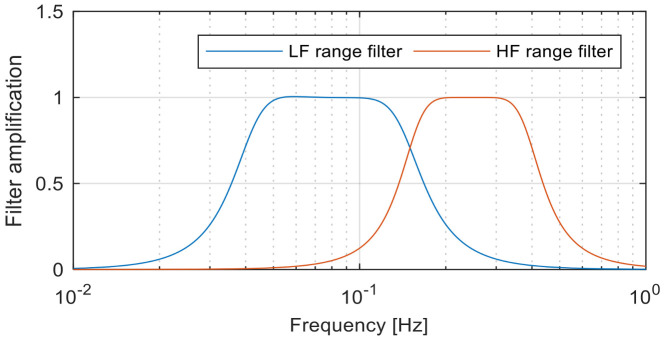
Frequency response of decomposition filters.

**Figure 8 sensors-20-07088-f008:**
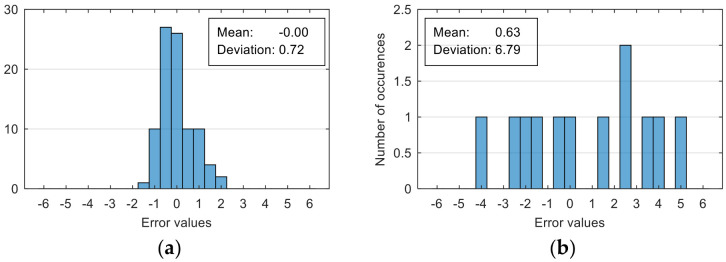
Estimation errors distribution for an improper data set split and 7 features (**a**) for identification subset; (**b**) for validation subset.

**Table 1 sensors-20-07088-t001:** Complete list of the 32 features extracted from the physiological signals.

Physiological Parameter	Domain	Features Type	Features
EDA	Time domain	(1) MAV	GSR_mav, SCL_mav, SCR_mav
		(2) MAR	GSR_mav, SCL_mav, SCR_mav
		(3) STD	GSR_mav, SCL_mav, SCR_mav
		(4) WL	GSR_mav, SCL_mav, SCR_mav
		(5) SSC	GSR_mav, SCL_mav, SCR_mav
		(6) WAMP	GSR_mav, SCL_mav, SCR_mav
	Event related	(1) NR	GSR_nr
		(2) AVRA	GSR_avra
		(3) MAXRA	GSR_maxra
	Frequency domain	(1) FMD	GSR_mav, SCL_mav, SCR_mav
		(2) FMN	GSR_mav, SCL_mav, SCR_mav
HR	Time domain	(1) MAV	HR_mav
		(2) STD	HR_std
	Frequency domain	(1) LFI	HR_lfi
		(2) HFI	HR_hfi
		(3) HLR	HR_hlr

**Table 2 sensors-20-07088-t002:** Experimental data organization for processing.

Subject Number	Data Window	ANX	Features
1	1	0….10	1, 2, 3, ……………….. 32
	2	0….10	1, 2, 3, ……………….. 32
	……	……	……
	15	0….10	……
……	……	……	……
7	1…15	0…10	1, 2, 3, ……………….. 32

**Table 3 sensors-20-07088-t003:** Ranking of features that showed the highest degree of correlation with anxiety.

Feature	Anxiety
SCL_mar	0.715 ***
GSR_mar	0.700 ***
GSR_wamp	0.372 ***
GSR_mav	0.350 ***
SCL_mav	0.345 ***
SCR_wamp	0.341 ***
SCR_fmd	−0.325 **
SCR_fmn	−0.319 **
SCL_wl	0.310 **
GSR_wl	0.309 **
SCL_std	0.305 **
SCR_wl	0.294 **
GSR_std	0.289 **
SCR_std	0.276 **
GSR_maxra	0.257 **
GSR_avra	0.256 **
SCR_mav	0.255 **
HR_lfi	0.212 *
GSR_fmn	−0.202 *
GSR_nr	0.192 *

*** *p* < 0.001, ** *p* < 0.01, * *p* < 0.05 (two-tailed).

**Table 4 sensors-20-07088-t004:** Distribution of the reported anxiety levels

Reported Anxiety Level	Occurrences in the Whole Data Set $NT_{K}$	Occurrences in the Identification Subset $NI_{K}=round\;\left(\frac{90}{105}NT_{K}\right)$	Occurrences in the Validation Subset $NV_{K}=round\;\left(\frac{15}{105}NT_{K}\right)$
0	41	35	6
1	16	14	2
2	12	10	2
3	9	8	1
4	7	6	1
5	4	3	1
6	4	3	1
7	2	2	0
8	5	4	1
9	2	2	0
10	3	3	0

**Table 5 sensors-20-07088-t005:** Estimation performance for different number of selected features

Number of Parameters Used	Parameter	Sum of Squared Estimation Errors		
		Whole Dataset	Identification Dataset	Validation Dataset
1	SCL_mar	3.827	3.782	4.092
2	HR_mav, GSR_mar	3.059	2.921	3.884
3	HR_mav, GSR_std, GSR_mar	2.663	2.519	3.526
4	HR_mav,GSR_mar, SCR_mar, SCR_fmd	2.041	2.050	1.983
5	HR_mav, SCR_mav, SCL_mar, GSR_wamp, SCR_wamp	1.656	1.674	1.546
6	HR_mav, HR_std, SCR_mav, SCL_mar, GSR_wamp, SCR_wamp	1.286	1.327	1.044
7	HR_mav, HR_std, SCR_mav, SCL_mar, SCR_mar, GSR_wamp, SCR_wamp	1.031	1.046	0.940

**Table 6 sensors-20-07088-t006:** Distribution of estimation errors and confusion matrices for levels 2 and 3—using one feature.

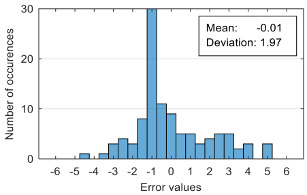					Real	Low	0.85	0.15	0.00
	Real	Low	0.96	0.04		Mild	0.59	0.23	0.18
		High	0.37	0.63		High	0.00	0.70	0.30
			Low	High			Low	Mild	High
		Estimated					Estimated		
	**Accuracy: 90.48%**				**Accuracy: 69.52%**				

**Table 7 sensors-20-07088-t007:** Distribution of estimation errors and confusion matrices for levels 2 and 3—using two features.

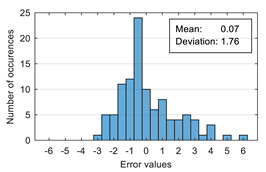					Real	Low	0.95	0.05	0.00
	Real	Low	0.98	0.02		Mild	0.29	0.65	0.06
		High	0.44	0.56		High	0.00	0.50	0.50
			Low	High			Low	Mild	High
		Estimated					Estimated		
	**Accuracy: 91.43%**				**Accuracy: 85.71%**				

**Table 8 sensors-20-07088-t008:** Distribution of estimation errors and confusion matrices for levels 2 and 3—using three features.

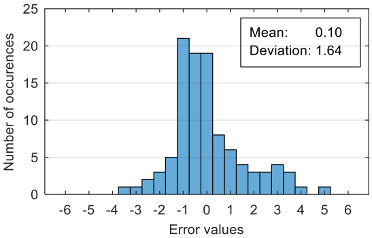					Real	Low	0.96	0.04	0.00
	Real	Low	0.99	0.01		Mild	0.41	0.47	0.12
		High	0.25	0.75		High	0.00	0.40	0.60
			Low	High			Low	Mild	High
		Estimated					Estimated		
	**Accuracy: 95.24%**				**Accuracy: 84.76%**				

**Table 9 sensors-20-07088-t009:** Distribution of estimation errors and confusion matrices for levels 2 and 3—using four features.

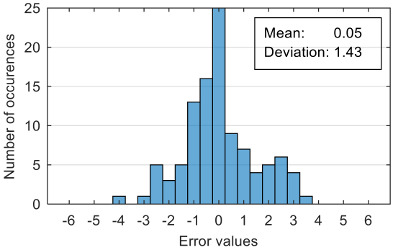					Real	Low	0.95	0.05	0.00
	Real	Low	0.99	0.01		Mild	0.18	0.76	0.16
		High	0.12	0.88		High	0.00	0.30	0.70
			Low	High			Low	Mild	High
		Estimated					Estimated		
	**Accuracy: 97.14%**				**Accuracy: 89.53%**				

**Table 10 sensors-20-07088-t010:** Distribution of estimation errors and confusion matrices for levels 2 and 3—using five features.

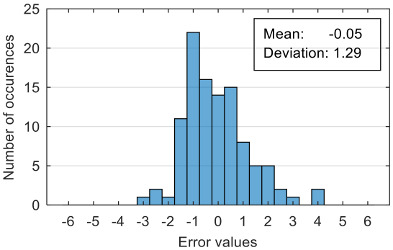					Real	Low	0.95	0.05	0.00
	Real	Low	0.98	0.02		Mild	0.35	0.53	0.12
		High	0.19	0.81		High	0.00	0.10	0.90
			Low	High			Low	Mild	High
		Estimated					Estimated		
	**Accuracy: 95.24%**				**Accuracy: 87.62%**				

**Table 11 sensors-20-07088-t011:** Distribution of estimation errors and confusion matrices for levels 2 and 3—using six features.

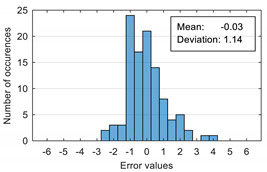					Real	Low	0.95	0.05	0.00
	Real	Low	0.97	0.03		Mild	0.18	0.76	0.06
		High	0.19	0.81		High	0.00	0.00	1.00
			Low	High			Low	Mild	High
		Estimated					Estimated		
	**Accuracy: 94.29%**				**Accuracy: 92.38%**				

**Table 12 sensors-20-07088-t012:** Distribution of estimation errors and confusion matrices for levels 2 and 3—using seven features.

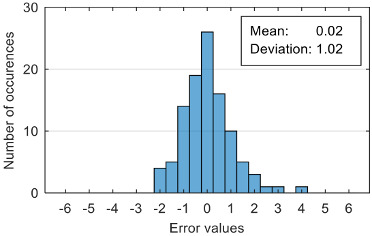					Real	Low	0.96	0.04	0.00
	Real	Low	0.99	0.01		Mild	0.23	0.65	0.12
		High	0.06	0.94		High	0.00	0.10	0.90
			Low	High			Low	Mild	High
		Estimated					Estimated		
	**Accuracy: 98.10%**				**Accuracy: 90.48%**				

**Table 13 sensors-20-07088-t013:** Comparison between different classifiers found in literature and our results.

Study	Application	Physiological Signals	Method	No. of Subjects	No. of Classes	Time Window	Accuracy
Zhang et.al. [102]	Public speaking anxiety	GSR	Neural Network	22	2 (high anxiety/calmness)	300 s	86.70%
					2 (high anxiety/ low anxiety		78.83%
Šalkevicius et al. [103]	Public speaking anxiety (VR exposure)	GSR, Blood Volume Pulse (BVP), skin temperature	SVM classifier	30	4 (low, mild, moderate, high)	20 s	86.30%
Sandulescu et al. [104]	Trier Social Stress Test (TSST)	GSR, BVP	SVM	5	2 (stressed, not stressed)	300 s (stress)/ 120 s (no stress)	73.26–83.08% (subject dependent)
Ihmig et al. [105]	Arachno-phobia (VR exposure)	GSR, HR, HRV, respiration	Bagged Trees	80	2 (low, high)	10 s	89.80%
						60 s	90.90%
					3 (low, medium, high)	10 s	74.40%
						60 s	73.40%
Chen et al. [106]	Driving stress	ECG, GSR, respiration	SVM	9	3 (low, medium, high)	100 s	89.70%
			ELM				89.20%
Liu et.al. [107]	Driving stress	GSR	Linear discriminant analysis	11	3 (low, medium, high)	300 s	81.82%
Sharma et al. [108]	Social Anxiety	GSR	MLP	31	2 (low, high)	120 s	85.70%
Our study	Fear of heights (VR exposure)	GSR, HRV	Regression model, threshold comparison	7	2 (low high)	5 s	94.29%
					3 (low, mild, high)		92.38%

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
