# Peer review of "Integrating Biosignals Measurement in Virtual Reality Environments for Anxiety Detection"

_sensors, 2020, doi:10.3390/s20247088_

Round 1

Reviewer 1 Report

This paper presents an approach to estimate the anxiety level using immersive virtual reality (based on the physiological signals). The paper is well written, the research appears to be sound, the results are exciting for their potential as a research tool, the study will be of interest to Sensors readers, and I believe that the paper merits publication in Sensors.

However, I do have some comments given as follows:

1). Only 7 subjects are involved in the experiments, which makes this method less convincing.

2). It is desirable to check the influence of different nonlinear functions.

3). The details of the feature extraction are missing. Moreover, it is interesting to know the feature importance for the classification task.

4). There are some stylistic issues in Reference, please double check it.

Some references can be helpful:

Fonseca, Eduardo, et al. "Acoustic scene classification by ensembling gradient boosting machine and convolutional neural networks." Virtanen T, Mesaros A, Heittola T, Diment A, Vincent E, Benetos E, Martinez B, editors. Detection and Classification of Acoustic Scenes and Events 2017 Workshop (DCASE2017); 2017 Nov 16; Munich, Germany. Tampere (Finland): Tampere University of Technology; 2017. p. 37-41.. Tampere University of Technology, 2017.

Xu, Kele, et al. "General audio tagging with ensembling convolutional neural networks and statistical features." The Journal of the Acoustical Society of America 145.6 (2019): EL521-EL527.

Author Response

Dear reviewer,

We have made corrections according to your comments and suggestions. Attached you can find the detailed responses to your concerns.

We are looking forward to hearing from you and we hope that you will find our paper suitable for publication.

Yours sincerely,

The authors

Reviewer 2 Report

The study is interesting and innovative. Some recommendations that I would like to make are the following:

  1. The problem statement regarding the protocols should be strengthen further.
  2. In my opinion the 3. Virtual reality in psychophysical experiments needs restructuring and not only the description of key words about VR.
  3. Cronbach’s alpha is not referred.
  4. The Discussion is an “extensive” presentation of this study’s results. Authors need to compare their results with some previous ones.
  5. Even if authors describe a case study, maybe recommendations based on specific VR design features or elements or protocols need to be more clearly presented.
  6. Maybe some implications for practice and research could be noticed (not appropriate due to the limited number of participants, n=7).
  7. Limitations of this study are not provided.

Author Response

(The authors gave the same response as above.)

Reviewer 3 Report

The paper pertains a timely topic, regarding a treatment via virtual reality that could become more and more used in the future. The paper is well written in terms of English language. However, some changes should be made. Please find below a point-by-point comment: 1. Introduction: it is well written and explains the rationale for this study. 2. Section “emotion and exposure therapy”: this section should be improved in terms of clarity and flow. For example, lines 70-73 introduce different theories, one after another. Ekman identified…Plutchik proposed… Izard presented… The authors should try to condensate the information in order to make the paragraph easier to follow. 3. Section “virtual reality in psychophysical experiments”: in line 143, the concepts of desensitization and habituation are introduced without explaining their meaning. The concept of habituation is only reported later in the text (line 211-212). Please explain these concepts when you first introduce them. 4. The other sections are OK, but the overall length should be limited. 5. In the results, repeat the parameters considered instead of stating “combinations of 1 to 7”. 6. Discussion: it must be improved, since it seems just a repetition of the results and does not compare your data with other experiments (even if not identical, at least similar, even if on other disorders). Literature is now rich of papers regarding virtual reality in anxious disorders and many other psychiatric conditions.

Author Response

(The authors gave the same response as above.)

Round 2

Reviewer 3 Report

The paper has bene improved in terms of clarity and flow. 

I do not have further suggestions.